# Potential Cycling Effects on Activities of Nickel-Mediated Benzyl Alcohol and Glycine Electrooxidation in Alkaline Solutions

**Yuyang Wei and Wenbin Zhang \***

College of Chemistry and Bioengineering, Yichun University, Yichun 336000, China; zhoumoli1@163.com
\* Correspondence: zhangwbycu@163.com

**Abstract:** Nickel electrodes under continuous potential cycling were applied for the electrooxidation of benzyl alcohol and glycine in KOH solutions, and their activities were measured and compared by cyclic voltammetry. It is shown that intrinsic activities of both reactions decrease with the increasing catalyst loadings, and a more significant decreasing trend was observed in glycine electrooxidation when compared to benzyl alcohol electrooxidation. These phenomena may be explained by an increasing of mass loading induced a decrease of the catalyst surface conductivity, structure changes of $Ni(OH)_2$ from $\alpha$-phase to $\beta$-phase, and the intercalation of glycine molecules into nickel hydroxide interlayers.

**Keywords:** nickel hydroxide; benzyl alcohol; glycine; potential cycling; electrocatalytic activity

## 1. Introduction

Nickel hydroxide-based materials have been extensively used in the electrooxidation of several small molecules [1,2]. Ni-(oxy) hydroxide-based materials such as nickel−iron oxyhydroxide (NiFeOOH) are promising earth-abundant catalysts for electrochemical water oxidation in basic media [3–5]. It was discovered in early 1971 that a series of amines and alcohols can be electrooxidized by nickel hydroxide [6,7]; for example, several electrooxidation reactions of benzyl alcohol in an alkaline medium were accomplished on different types of nickel hydroxide electrodes [8,9]. Glycine, as a biosynthetic intermediate of all purines and porphyrins, can be electrooxidized by $Ni(OH)_2$ as well [10]. High-catalytic activity has been obtained in nickel manganese oxide with spinel nanostructure-mediated electrooxidation reaction toward glycine in alkaline electrolytes [11]. Due to its potential applications in wastewater treatment, fuel cells, hydrogen production, and urea electrolysis, using nickel hydroxide-based materials as electrocatalysts has attracted much attention [12–14]. Besides the above-mentioned, nickel hydroxide-based catalysts have been used in electrocatalytic reactions related to formaldehyde [15] and sulfide [16].

In order to optimize both intrinsic activity and the number of active sites for particular electrolysis applications, how intrinsic activity changes as a function of the catalyst loading should be understood thoroughly. It was observed that specific electrooxidation activity related to glycerol decreases with increasing Au loading [17], and this finding can be explained by the increasing gold oxide inhibition, which is caused by the accumulation of oxidation products within the electrocatalytic layers. When nickel hydroxide was used as a cocatalyst for bismuth vanadate and iron vanadate photoanodes for water oxidation, the large loadings of $Ni(OH)_2$ induce a drastic decrease in the photoelectrocatalytic activity because of the charge recombination at the semiconductor/$Ni(OH)_2$ interface [18–20].

Boettcher et al. discovered that in studies of oxygen-evolution reactions [21], turnover frequency of pulse-deposited Ni–Fe (oxy)hydroxide films increase with increasing loadings, whereas continuously

electrochemically deposited films show the opposite relationship. It is generally accepted that $Ni(OH)_2$ forms electrochemically on Ni electrodes in alkaline media [22,23]. Potential cycling is one of the most versatile and convenient techniques that can be used to generate hydrous oxides with a real-time determination of active centers and subsequent electrocatalytic activities. Changes of structure, amount, and other properties of nickel hydroxide induced by potential cycling have been addressed by atomic-force microscopy (AFM), Raman spectroscopy, X-Ray Photoelectron Spectroscopy (XPS), ellipsometry, and gravimetry with an electrochemical quartz-crystal microbalance [23–29]. For example, in situ AFM with the tip fixed has been used to monitor the changes of nickel hydroxide-film thickness during the potential cycling process [21]; in situ Raman spectroscopy observations indicate that aging of γ-NiOOH film in KOH electrolyte by electrochemical cycling produces a β-NiOOH film [25].

To the best of our knowledge, the influence of potential cycling on the activity of nickel-based electrocatalytic reactions is still obscure, and the relationship between the intrinsic activity and catalyst loading needs to be addressed. Herein, electrooxidation reactions of benzyl alcohol and glycine were performed by an electrodeposited nickel electrode in 1 mol $L^{-1}$ KOH. The effects of increased nickel hydroxide loadings and other changes induced by potential cycling on electrocatalytic activities were investigated by cyclic voltammetry. It should be noted that the reason for choice of these two substrates was because of their chemical structural simplicity, and conclusions from them can be easily applied to other alcohols or amino acids.

## 2. Results and Discussion

Cyclic voltammograms recorded during the growth of $Ni(OH)_2$ film on electrodeposited Ni particles in 1.0 mol $L^{-1}$ KOH are presented in Figure 1; the amount of electrodeposited Ni measured by chronoamperometry is $2.6 \times 10^{-7}$ mol $cm^{-2}$. It is calculated using the equation $\Gamma_{Ni} = Q_{Ni}/2FA$, in which $Q_{Ni}$ can be calculated from the integration of current–time curve, F is the Faraday's constant, and A is the geometric surface area of the copper electrode. The growth of the film is evident from the anodic- and cathodic-peak current density, which becomes more developed with an increasing number of potential cycles. It can be assumed that the reduction peak represents the conversion from nickel hydroxide to nickel oxyhydroxide, and the oxidation peak is caused by the oxidation of Ni (0) to $Ni(OH)_2$ and the oxidation of $Ni(OH)_2$ to NiOOH. Since the cathodic charge is almost equal to the anodic charge, a conclusion could be drawn that peak current contributed by the generation of $Ni(OH)_2$ from Ni (0) in the oxidation-peak current is negligible. Here, cathodic charge was used to calculate the amount of $Ni(OH)_2$. The variation of $\Gamma_{Ni(OH)2}$ with the number of potential cycles is presented in the inset of Figure 1. It is obvious that nickel hydroxide loadings on the copper electrode increase with the potential cycling, and the surface amount of nickel hydroxide increases to around $9 \times 10^{-8}$ mol $cm^{-2}$ within 80 cycles at a scan rate of 10 mV/s.

Different loadings of nickel hydroxide were generated by potential cycling of a nickel-film-modified electrode in 1 mol $L^{-1}$ KOH. Nickel hydroxide loadings were calculated by integrating the $Ni(OH)_2$/NiOOH oxidation wave, assuming one electron per Ni oxidized, the corresponding equation is $\Gamma_{Ni(OH)2} = Q_{Ni(OH)2}/FA$, where $Q_{Ni(OH)2}$ is the cathodic-voltammetric charge corresponding to the oxidation process. The electrochemical activity of the catalyst upon different potential cycles was investigated for the benzyl alcohol electrooxidation in 1 mol $L^{-1}$ KOH/0.02 mol $L^{-1}$ benzyl alcohol solution at 298 K by cyclic voltammetry, with the addition of benzyl alcohol, the height of the anodic peak increases with the cathodic peak decreasing, the characteristic of a classical electrocatalytic mechanism (EC′) was observed in Figure 2a. A general mechanism for the oxidation of benzyl alcohol was proposed and shown in Scheme 1, based on previous work [7]. Meanwhile, an increasing anodic/cathodic current density was discovered with the increasing potential cycling number, when the anodic-peak current density was 3 mA $cm^{-2}$ and with a catalyst loading of ~$3 \times 10^{-8}$ mol $cm^{-2}$, while it was 5.5 mA $cm^{-2}$ when the loading increased to ~$1 \times 10^{-7}$ mol $cm^{-2}$ (Figure 2b inset). It is reasonable to conclude that more benzyl alcohol molecules can be electrochemically oxidized with the increasing concentration of nickel hydroxide. To evaluate whether intrinsic activity was independent of catalyst

loading or not, the relative anodic-peak current densities $j_c/j_b$ ($j_c$ and $j_b$ represent the anodic-peak current density in the presence or absence of benzyl alcohol, respectively) as a function of nickel hydroxide loadings was demonstrated in Figure 2b. It is obvious that intrinsic activity (the catalytic ability of per nickel hydroxide molecule) relative to benzyl alcohol decreases with the increasing total nickel hydroxide amount during potential cycling.

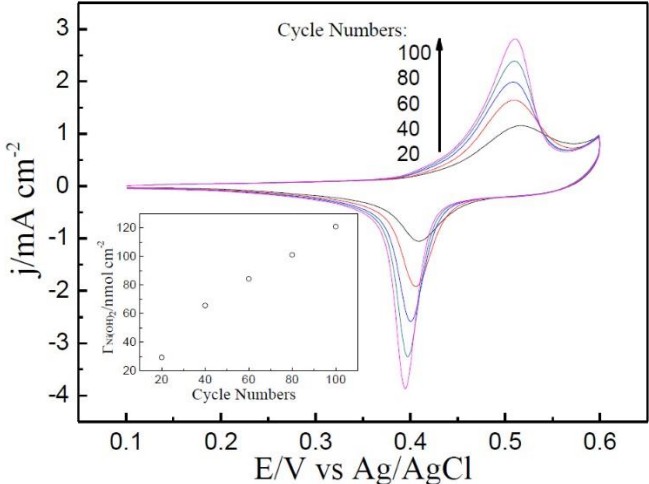

**Figure 1.** Cyclic voltammograms of the Ni electrode in 1 mol $L^{-1}$ KOH under different potential cycles, inset: loadings of nickel hydroxide as a function of oxide-growth-cycle numbers.

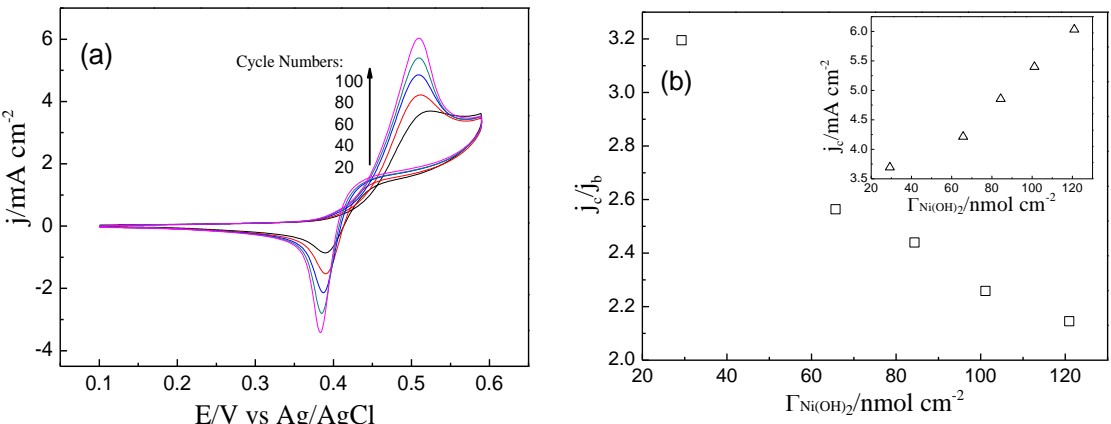

**Figure 2.** (**a**) Cyclic voltammograms of Ni in 1 mol $L^{-1}$ KOH/20 mmol $L^{-1}$ benzyl alcohol after submitting to 1 mol $L^{-1}$ KOH upon different numbers of potential cycling, (**b**) relative anodic-peak current density and anodic-peak current density (inset) as a function of nickel hydroxide loadings for benzyl alcohol electrooxidation.

Nickel hydroxides formed by potential cycling and continuously precipitated on the outer surface of the film, based on Tucceri's work [30], might result in a reduced conductivity of the electrode with the increasing cycling numbers. It should also be noted that the potential cycling approach involves a slow conversion of $\alpha$-Ni(OH)$_2$ to a more thermodynamically favored $\beta$-phase of Ni(OH)$_2$ [22–24]. Compared to $\alpha$-Ni(OH)$_2$/$\gamma$-NiOOH, a more ordered $\beta$-Ni(OH)$_2$/$\beta$-NiOOH displays a less accessible ion–solvent intercalation during electrocatalysis. Moreover, according to the studies of Nocera et al. in the nickel-based, oxygen-evolving catalyst $\gamma$-NiOOH [27], including a higher oxidation state of Ni than $\beta$-NiOOH, can favor the formation of hydroperoxy (OOH) species (key intermediates in the alcohol electrooxidation [22]), which make it more effective as a catalyst in the electrooxidation reaction toward benzyl alcohol.

Opposite behavior of electrocatalytic-peak current densities as a function of potential cycle numbers (loadings of nickel hydroxide) was observed in the study of glycine electrooxidation (Figure 3a,b inset). A more significant decrease of intrinsic activities dependent on nickel hydroxide loadings was discovered in glycine electrooxidation when compared with benzyl alcohol electrooxidation. If these two relationships can be treated approximately as linear relations, for the case of glycine electrooxidation, the slope is around $-0.98$ nmol$^{-1}$ cm$^2$, while it is around $-0.011$ nmol$^{-1}$ cm$^2$ for benzyl alcohol electrooxidation. Based on the studies of Fleischmann et al. [7], a possible reaction mechanism for Ni(OH)$_2$-mediated glycine electrooxidation has been shown in Scheme 2.

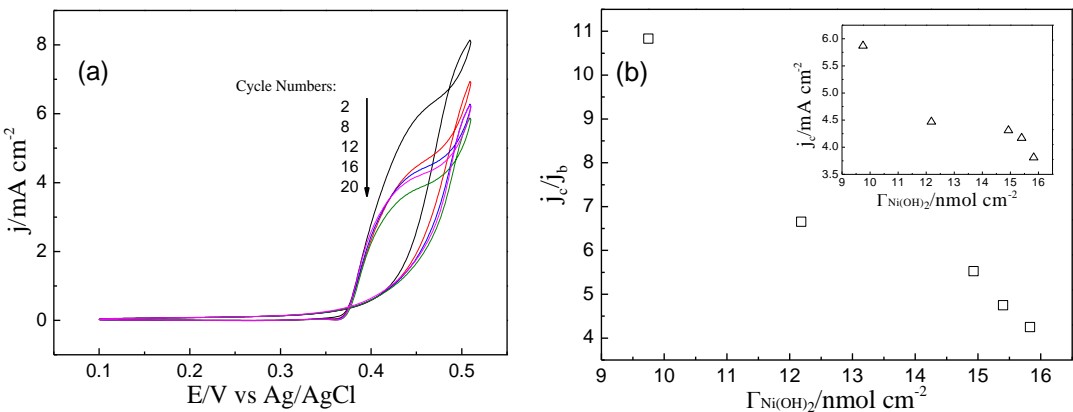

$$Ni(OH)_2 + OH^- \rightleftharpoons NiOOH + H_2O + e^-$$

**Scheme 1.** Reaction mechanism of benzyl alcohol electrooxidation mediated by Ni(OH)$_2$.

**Figure 3.** (**a**) Cyclic voltammograms of Ni in 1 mol L$^{-1}$ KOH in the presence of 20 mmol L$^{-1}$ glycine under different cycle numbers, (**b**) relative anodic-peak current density and anodic-peak current density (inset) as a function of nickel hydroxide loadings for glycine electrooxidation.

$$Ni(OH)_2 + OH^- \rightleftharpoons NiOOH + H_2O + e^-$$

$$NiOOH + OH^- \longrightarrow NiOO^- + H_2O$$

**Scheme 2.** Possible reaction scheme for the catalytic oxidation of glycine by $Ni(OH)_2$.

Yodhida et al. discovered that a glycine molecule could combine with the $NiO_6$ clusters by binding with $NH_2$ and COOH groups, leading to the formation of NiOOH–glycine complex [31], therefore, it is reasonable to postulate that upon potential cycling, glycine molecules will transport into interlayers of NiOOH, resulting in the formation of a similar NiOOH–glycine complex with the diminishing of glycine-surface amount. It can be suggested that besides the electronic conductivity decreasing and structure changes, incorporation of glycine into the catalyst should also be responsible for the reduction of the catalytic activity in glycine electrooxidation.

## 3. Materials and Methods

Copper-electrode (d = 3.0 mm) modification with nickel films was achieved by electrodeposition in 1 mol $L^{-1}$ $NiSO_4$ + 0.5 mol $L^{-1}$ $H_3BO_3$ electrolyte under constant potential −1.25 V (vs. Ag/AgCl, sat. KCl) for 60 s. Ninety-nine and nine-tenths percent nickel sulfate hexahydrate, 99% benzyl alcohol, 99% nickel nitrate hexahydrate, 99.99% potassium hydroxide, and 99.5% boric acid were purchased from Aladdin, and biotechnology-grade glycine was obtained from Macklin. All chemicals were used as received without further purification, and all solutions were prepared using ultrapure water (18.2 MΩ cm), which was made from a purifier ultrapure water system. Cyclic voltammetry was performed using an CHI440C electrochemical workstation (Chenhua, China), and the scan rate used was 10 mV/s. An electrochemical cell with a three-electrode configuration was used here, a platinum plate and an Ag/AgCl/KCl (sat.) were employed as counter and reference electrodes, respectively. All electrodes were obtained from Gaossunion (Wuhan, China).

## 4. Conclusions

In order to investigate the influence of potential cycling on catalytic abilities of nickel hydroxide-mediated small-molecule electrooxidation reactions, benzyl alcohol and glycine electrooxidation was achieved and characterized by cyclic voltammetry; different loadings of nickel hydroxide onto nickel films were obtained by using potential cycling as a simple approach. It was shown that for benzyl alcohol electrooxidation, intrinsic activities decrease with the increased loadings, this decreasing trend may be caused by the increase of surface resistance and structure variation of the catalyst. A more evident decreasing trend was discovered in glycine electrooxidation, and besides the two reasons noted above, this is also caused by the intercalation of glycine to interlayer with nickel oxyhydroxide. Further explanations for the observed phenomena are still under investigation. This work is a reminder that when looking into nickel-mediated, small-molecule electrocatalytic reactions in alkaline solutions, the potential cycling effect should not be overlooked.

**Author Contributions:** Y.W. performed the experiments; W.Z. contributed to the conceptualization and Writing. All authors have read and agreed to the published version of the manuscript.

**Funding:** This research was funded by the National Natural Science Foundation of China under grant number 21962020.

**Conflicts of Interest:** The authors declare no conflict of interest.

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
