# Peer review of "Potential Cycling Effects on Activities of Nickel-Mediated Benzyl Alcohol and Glycine Electrooxidation in Alkaline Solutions"

_catalysts, doi:10.3390/catal10010119_

Round 1

Reviewer 1 Report

This ms contains correct results on the Ni-mediated electrooxidation of benzylalcohol and glycine, however, the whole ms may be rather boring for the majority of the readers of the high impact journal, Catalysts. I do not think that electrochemistry belongs to the primary scope of Catalysts. Au-s should select a more specialized journal. It was surprising that the au-s not even presented the fate of the starting substrates by simple schemes. This Referee beleives that Catalysts is somewhat connected to organic chemistry.

Author Response

Thanks for the comment, I believe electro-catalyst is an important of catalyst, considering its importance in energy, environment, organic synthesis and other areas. Reaction schemes for benzyl alcohol and glycine electrooxidation reactions have been added in catalysts-688623R1. 

Reviewer 2 Report

I suggest the manuscript be considered for publication after minor review in terms of language and choice of words. 

Author Response

Thanks for this comment, language has been polished again and marked in red.

Reviewer 3 Report

The revised manuscript has addressed all issued from the original submission and can be published now.

Author Response

Thank you for this comment!

Round 2

Reviewer 1 Report

The prsentation became better and the chemistry can now be seen.

At the sam tim, I feel tat the subjct isnot too inteesting and of bordeline importanc.

However, if the Editors beleive that the subject belongs to th scop of Catalyst, the ms may be acceptable.